# Problems Associated with Co-Infection by Multidrug-Resistant *Klebsiella pneumoniae* in COVID-19 Patients: A Review

**DOI:** 10.3390/healthcare10122412

**Published:** 2022-11-30

**Authors:** Reham Omar Yahya

**Affiliations:** 1Basic Sciences Department, College of Sciences and Health Professions, King Saud Bin Abdulaziz University for Health Sciences, Riyadh 11671, Saudi Arabia; yahyar@ksau-hs.edu.sa; 2King Abduallah International Medical Research Center, Riyadh 11481, Saudi Arabia

**Keywords:** multidrug-resistant, *Klebsiella pneumoniae*, COVID-19, co-infection

## Abstract

To date, coronavirus disease 2019 (COVID-19) and its variants have been reported as a novel public health concern threatening us worldwide. The presence of *Klebsiella pneumoniae* in COVID-19-infected patients is a major problem due to its resistance to multiple antibiotics, and it can possibly make the management of COVID-19 in patients more problematic. The impact of co-infection by *K. pneumoniae* on COVID-19 patients was explored in the current review. The spread of *K. pneumoniae* as a co-infection among critically ill COVID-19 patients, particularly throughout hospitalization, was identified and recorded via numerous reports. Alarmingly, the extensive application of antibiotics in the initial diagnosis of COVID-19 infection may reduce bacterial co-infection, but it increases the antibiotic resistance of bacteria such as the strains of *K. pneumoniae.* The correct detection of multidrug-resistant *K. pneumoniae* can offer a supportive reference for the diagnosis and therapeutic management of COVID-19 patients. Furthermore, the prevention and control of *K. pneumoniae* are required to minimize the risk of COVID-19. The aim of the present review is, therefore, to report on the virulence factors of the *K. pneumonia* genotypes, the drug resistance of *K. pneumonia*, and the impact of *K. pneumoniae* co-infection with COVID-19 on patients through a study of the published scientific papers, reports, and case studies.

## 1. Introduction

The coronavirus disease 2019 (COVID-19) pandemic has quickly grown across all developed and non-developed countries, representing an unparalleled challenge for therapeutic organizations. At the beginning of the pandemic outbreak, there were differences in the estimated mortality rates, which cannot be fully explained. During this pandemic, there has been a knowledge gap regarding the mechanisms of COVID-19 co-infection by microbial pathogens. Numerous reports have shown that microbial co-infections in COVID-19 patients are still emerging. The coronaviruses (CoVs) are a large group of coated RNA viruses, some of which cause human health imbalances, such as Middle East respiratory syndrome (MERS), severe acute respiratory syndrome (SARS), and others that invade certain animals [1]. A respiratory sickness caused by the current coronavirus disease 2019 (COVID-19), caused in turn by SARS-CoV-2, became infamous at the end of 2019, starting in Wuhan, Hubei Territory, China [1]. The current episodes of CoV disease remind us that CoVs are still serious and represent an open danger to health worldwide. SARS-CoV-2 is the seventh CoV identified to infect humans [1]. As already known, it belongs to the subgenus Sarbecovirus of the Coronaviridae family. There are some similarities between SARS-CoV-2 and other previously detected coronaviruses (SARS-CoV-2 and MERS-CoV), but it is distinct from them in certain aspects, such as its transmission rate, origin, and mortality rate [1]. Based on the severity of the presenting illness, COVID-19 was classified into five distinct severities (Figure 1). However, it was different between reports conducted in different locations, and there is still some doubt about the percentage of asymptomatic infections [2]. Furthermore, the description of asymptomatic infections may differ within the published medicinal reports, depending on which definite symptoms were judged. For instance, according to the study by Sakurai et al. [2], at the time of diagnosis, 58% of the 712 individuals confirmed to have COVID-19 were asymptomatic. Ma et al. [3] mentioned that individuals with asymptomatic infections represented in 40.50% of cases. During the outbreak of the Omicron variant in South Africa, an important investigation assessed that 31% of the confirmed cases were asymptomatic. Therefore, according to Ma et al. [3], discovering asymptomatic infection is essential, particularly in regions that have effectively controlled SARS-CoV-2.

According to available data from the World Health Organization, from the beginning of COVID-19 outbreaks to 2022, there were more than 574 million confirmed cases and more than 6.3 million deaths associated with SARS-CoV-2 and its variants worldwide. *Klebsiella pneumoniae*, appearing in the bacillus form, was isolated from the respiratory system of humans infected with pneumonia, and examined by Carl Friedlander in 1882. Initially termed Friedlander’s bacillus, it was not until 1886 that the microbe earned the title *Klebsiella*. The familiar pathogen *K. pneumoniae*, recognized as an enteric bacterium, has a place among the family of Enterobacteriaceae, characterized and illustrated as non-motile and Gram-negative (Gr-ve). *K. pneumonia* may be a common tenant of the digestive tract microbiome in humans and other creatures. It is a familiar hospital-associated causative agent of the disease, accounting for approximately one-third of Gr-ve bacteria contaminations in general.

Even though the world is working to solve the outbreak of COVID-19, the appropriate management of and immunization against widespread COVID-19 continue to take place vaguely across the globe. Furthermore, co-infection by SARS-CoV-2 and other causative breathing diseases has become another genuine issue for the treatment of COVID-19 patients. Generally, microbial co-infection increases the risk of serious illness. COVID-19 is recognized as a modern microbial infection. The phenomenon of microbial co-infections presents one of the greatest recovery concerns, particularly viral–bacterial co-infections, which are now occurring in expanded proportions and increasing the threat of mortality. Recently, scientific papers documented the existence of various microbes as co-pathogens in patients infected by COVID-19 (Figure 2), such as *K. pneumoniae*, *Staphylococcus aureus, Haemophilus influenzae*, *Pseudomonas aeruginosa,* and *Legionella pneumophilia*, alongside other pneumonia species, including *Mycoplasma pneumoniae* and *Streptococcus pneumoniae.*

In a newly published scientific paper, 8.2% of the COVID-19 patients were found to be infected with various pulmonary bacteria, and 64.8% of the main detected bacteria were Gram-negative bacteria. Among the detected bacteria, *Acinetobacter baumannii*, *K. pneumoniae*, and *P. aeruginosa* represented 31.9%, 19.8%, and 8.8% of cases, respectively [4]. Based on the published papers, Santos et al. [5] mentioned that a number of COVID-19 patients (1394 cases) were co-infected with the pathogenic bacteria, but with different percentages, including *Enterobacter* spp. (35%), *S. aureus* (27%), *Klebsiella* spp. (21%), *Escherichia coli* (13%), coagulase-negative *Staphylococcus* (16%), and *P. aeruginosa* (3%).

Not only bacteria, but also numerous other microbes, including rhinovirus, flu infection, other CoVs, *para*-influenza, *meta*-influenza, and human orthopneumovirus, have been described as conceivable co-infectious agents in COVID-19 patients [6]. Occasionally, co-infections by fungi have also been registered among patients infected by SARS-CoV-2, and the detailed infectious agents included unicellular and filamentous fungi, such as *Cryptococcus* spp., *Candida* spp. and *Aspergillus* spp. [7].

According to a published paper, the *Klebsiella* species is considered among the highest ten bacteria that cause nosocomial infections, and it is considered one of the foremost familiar infectious agents in the intensive therapy unit (ITU) [8]. The facts indicate that among these diseases, *K. pneumoniae* is responsible for a critical number of infections in the urinary tract, lungs, and delicate tissue diseases. The foremost means of the spread of *Klebsiella* are through the alimentary canal and the hands of the clinic team [9]. Cases of extreme infection in the ITU were co-infected by *K. pneumoniae*, a problem accompanied by the rise of *K. pneumoniae* strains that resist numerous narrow-spectrum antibiotics as well as broad-spectrum antibiotics. Generally, it is critical to constrain the hazards of the infection and the spread of the disease-causing agents. To date, insufficient studies have examined the bacterial co-infections of COVID-19 patients in Saudi Arabia, as well as other countries. Therefore, the present review outlined the drug resistance and virulence factors of *K. pneumonia* and provided a current survey of *K. pneumoniae* co-infection in COVID-19 patients.

## 2. Materials and Methods

The data reported and collected in the current review were dependent on the guidelines for Preferred Reporting Items for Systematic Reviews and Meta-Analyses (PRISMA). PRISMA focuses on reviews which measure the influences of interventions.

### 2.1. Inclusion Criteria

The following criteria were the bases of the studies included in the current review, including the articles of the original studies.

### 2.2. Exclusion Criteria

Letters to the editor, conference articles, commentaries, and viewpoint studies were excluded, and non-English-language scientific papers were also excluded.

### 2.3. Search Strategy

A search via electronic sources was employed to identify the numerous scientific papers written in English that are available on the MEDLINE website through search terms such as medico-legal litigation COVID-19, liability claims COVID-19, medical liability pandemic, and malpractice SARS-CoV-2. Additionally, the databases on the Web of Science, PubMed, Scopus, CINAHL, and Cochrane were searched under “*Klebsiella pneumoniae*” and “Resistant”. The goal of the present review is to focus on the documents, articles, reports, and study cases that contain useful information on the virulence factors of *K. pneumoniae* and co-infections by *Klebsiella pneumoniae* in COVID-19 infections. Some other search terms were employed in the current review, such as antibacterial drug resistance or resistance to antibiotics, drug resistance, antibiotic susceptibility, COVID-19, severe acute respiratory syndrome coronavirus 2, SARS-CoV-2 Infections, COVID-19 pandemic, coronavirus disease 19, and severe acute respiratory syndrome coronavirus 2.

## 3. Collected Data and Discussion

### 3.1. Virulence Factors and Their Relations to Klebsiella pneumoniae

*K. pneumoniae*, according to Bazaid et al. [10], was the most commonly detected species in ICU and non-ICU patients infected with SARS-CoV-2. Different virulence factors were detected in *K. pneumoniae*, leading to higher mortality [10].

The appearance of bacterium virulence presented via a broad cluster of components that can lead to the appearance of disease, in addition to antibiotic resistance. The most significant virulence factor was represented by the polysaccharide capsule (PSC) of the microbes. PSC permits and helps the microbes to avoid opsonophagocytosis and serum death in humans. Up to the present time, 77 diverse capsular types have been considered, and those capsules lacking the *Klebsiella* species have a tendency to be somewhat destructive. Lipopolysaccharides (LPS) are considered a critical destructive factor and surround the bacterial outer wall. The detection of LPS releases a fiery cascade in the half-life form and appears to be a main guilty party of the sequela in sepsis and infection shock. Additionally, fimbriae, which are recognized as virulent, permit the pathogen to connect itself to the cells of the host.

Additionally, there are other destructive factors, such as siderophores, that are required by the pathogen to cause infection in the host. Siderophores obtain iron, which exists the host, showing a vital function in the propagation process of the pathogen [11,12]. *K. pneumoniae* features a pathogenicity related to an assortment of harmful components (including the improvement of the capsule, lipopolysaccharide, hyper-mucoviscosity, and press-securing framework), all of which help the pathogen to overcome the human host’s natural resistance and maintain the infection in the host [13]. Severe community-acquired infections were detected among moderately healthy individuals with hypervirulent *K. pneumoniae* (HvKp). Russo and Marr [14] demonstrated the occurrence of virulence plasmids in the isolates of HvKp that carry virulence genes. HvKp, compared with the traditional *K. pneumonia*, is characterized by higher frequency, and it causes dispersed infections in the lungs, liver, eyes, and central nervous system. In spite of the fact that infections via HvKp have been described primarily in HvKp-endemic regions, such as eastern Asia, over time, scattered cases have been progressively described around the world [15].

### 3.2. Drug Resistance and Resistant Genotypes of Pathogenic K. pneumoniae

Regarding the risk factors, it is vital to observe that, occasionally, antibiotics are self-administered by persons or prescribed by a medical doctor to avoid infection by bacteria, even in cases where there is no confirmed infection or lab-based confirmation, in addition to the inappropriate use of the antibiotics in hospitals. The broad utilization of antibiotics for more than 80 years to prevent the outbreak of human microbial pathogens has created a familiar problem known as antibiotic resistance. Microbial resistance to antibiotics is presently recognized as a worldwide emergency in regard to present-day medication. Data regarding distinctive sorts of micro-organisms and their resistance could help to direct doctors, infection control actions, and approach producers in different nations and locales in making evidence-based choices so as to overcome the problem of resistance to antibiotics [16]. In the 1980s, a few strains of *K. pneumoniae* were observed to be resistant to many beta-lactam antibiotics. This set of antibiotics, which are among the foremost commonly endorsed classes of pharmaceuticals, include penicillin and its subsidiaries.

The resistance to antibiotics of *K. pneumoniae* has significantly expanded, making it an imperative danger around the world. *K. pneumoniae*-CRE has a very high resistance profile to most antibiotic categories (Figure 3). Consequently, it is very challenging to treat the disease. A number of antibiotics resistance mechanisms have been documented in the published scientific papers, including drug inactivation, the modification of drug binding sites, reduced intracellular drug accumulation, and biofilm formation. Thus, *K. pneumoniae* modifies the structure of the target sites to inhibit or block the efficient binding sites [17].

Other different mechanisms were also documented, including transformation, conjugation, and transduction, wherein susceptible *K. pneumoniae* strains can attain resistance genes with transposons that ultimately support diverse resistance genes so as to join with the plasmids or chromosomes of the host [18].

Various outer membrane proteins (Omps), such as OmpA, OmpF, and OmpC, were reported as contents of the cell wall of *K. pneumoniae*. Via diffusion, some antibiotics can cross through these Omps. Other cell surfaces are covered by these proteins, contributing to the preservation of the cell structure and regulation of antibiotics and nutrition transport [19].

To date, numerous *K. pneumoniae* strains have developed a great diversity of *β*-lactamase, carbapenemase, and metallo-*β*-lactamase enzymes, which can break the construction of antibiotics related to *β*-lactam, such as carbapenems, penicillins, and cephalosporins. Carbapenemase- and new-Delhi-metallo-*β*-lactamase-producing *K. pneumoniae* exhibited resistance not only to drugs associated with *β*-lactam antibiotics but also to aminoglycoside and quinolone antibiotics [20].

Later reports highlighted the development of multidrug-resistant *K. pneumoniae* strains which appear to have resistance to colistin, a last-line antibiotic, emerging from the mutational inactivation of the mgrB regulatory gene. Hence, studies of colistin-resistant *Klebsiella* isolates are increasing in number. According to a recent study by Narimisa et al. [21], there was an increment in the level of colistin resistance by *K. pneumoniae* during the period of 2013 to 2021, which may be attributed to the increased utilization of this antibiotic in recent years. A strong relationship was observed between the increased resistance to colistin and carbapenem-producing *K. pneumoniae* [21].

Imipenem belongs to the carbapenems, and it was recognized as the first carbapenem antibiotic according to Baumgartner and Glauser [22]. Imipenem was applied to the treatment of infections resulting from *K. pneumoniae* back in 1983, but after 2 years, imipenem-resistant strains of *K. pneumoniae* were discovered [23].

Antibiotics among the aminoglycoside class were used for the treatment of different human infections and were highly effective, until they were displaced by other antibiotics, such as third-generation cephalosporins, carbapenems, and fluoroquinolones [24]. Humanity did not benefit from the existence of this class of antibiotics, as the *K. pneumoniae* strains were able to weaken the efficiency of that class through special mechanisms, including antibiotic modification enzymes with diverse activities, such as acetylation and adenylation. The diminished utilization of aminoglycosides moderated the advancement of modern resistance components until the discovery of 16S rRNA methylase [25]. Since 2005, tigecycline (firstly glycylcycline) has been utilized for infections caused by *K. pneumoniae* after it was discovered that it appears to evade most of the identified resistance mechanisms against antibiotics belonging to the tetracyclines group and was shown to evade the major identified resistance tools against these groups [26].

It is known that tigecycline is characterized as a broad-spectrum antibiotic, and Golan [27] observed that tigecycline was a promising compound against antibiotic-resistant Enterobacteriaceae, creating extended-spectrum beta-lactamases (ESBLs) such as *K. pneumoniae*. However, after a short period of application, numerous multi-drug-resistant (MDR) *K. pneumoniae* strains were isolated from infected persons taking tigecycline and decreased their susceptibility toward these antibiotics. According to Du et al. [28], the mechanism of tigecycline resistance in *K. pneumoniae* has still not been clearly clarified; however, the tetA gene may be responsible for its antibiotic resistance. Therefore, Du et al. [28] confirmed that all the clinical strains of *K. pneumoniae* that carry *tetA* are associated with the failure of treatments using tigecycline.

A number of resistance genes were recognized in seven strains of *K. pneumoniae* (three tigecycline-resistant and four colistin-resistant strains). The sequencing of the gene mutations [29] from the resistant *K. pneumoniae* strains showed that the mutations of the *ramR* and *lon* genes were the probable explanations for tigecycline resistance, while colistin resistance may be due to the *pmrB*, *phoQ*, and *mgrB* genes. Resistance genes were recently studied in *K. pneumoniae* isolates, and the authors showed that blaCTX-M-1 was the most common *β*-lactam resistance gene, representing 91%, followed by 76.1% for blaTEM, 68.7% for blaSHV, 29.9% for blaOXA-1, 14.9% for blaGES, 11.9 % for blaCTX-M-9, and 4.5% for blaCTX-M-2. Meanwhile, blaKPC was the most common carbapenemase resistance gene, representing 55.2%, followed by blaKPC representing 55.2%, blaIMP representing 28.4%, blaVIM representing 14.9%, blaNDM-1 representing 13.4%, and blaOXA-48 representing 10.4%. Moreover, virulence-associated genes such as fimH, representing 71.6%, ugeF representing 58.2%, wabG representing 56.7%, ureA representing 47.8%, and kfuBC representing 28.4% were also noticed [8].

Efflux-mediated resistance mechanisms were investigated in *K. pneumonia* via the efflux pump genes’ expression. For instance, acrA, acrB, and tolC were recorded as a mechanism of tigecycline-resistant *K. pneumonia* [30]. As reported by Moya and Maicas [31], the modifications in DNA gyrase in specific regions (gyrA and gyrB genes) and topoisomerase IV in specific regions (parC and parE genes) lead to resistance to fluoroquinolone. The resistance of *K. pneumonia* to diverse classes of antibiotics has been described in numerous nations. In the Aljouf region of Saudi Arabia, Bandy and Almaeen [32] found that 46% of the *K. pneumoniae* isolates from circulatory system infections were carbapenemase producers. Additionally, in Saudi Arabia, Farah et al. [33] found that the resistance rates of the *K. pneumoniae* isolates appeared to be more than 90% and above than 80% with respect to the first generation of the antibiotics associated with cephalosporins and to the second and fourth generations, respectively. Consequently, the recent results regarding *K. pneumoniae* resistance support the earlier results in Saudi Arabia. The predominance and patterns of *K. pneumonia* antibiotic resistance were studied by Al-Zalabani et al. [34] in Saudi Arabia over the course of 5 years, during which the resistance levels to *β*-lactams, in addition to a rise in resistance to carbapenems and to other options for treatment, such as tigecycline and colistin, were increased. A large percentage, representing up to 100% of the *K. pneumoniae* isolates, were resistant to diverse antibiotics, including cefepime, cefotaxime, ceftriaxone, ampicillin, and ceftazidime, while the amikacin and imipenem antibiotics were effective against 8.7% of the *K. pneumoniae* isolates in Saudi Arabia [35]. Generally, *K. pneumoniae* is currently prevalent in healthcare locations because of carbapenemase production and other resistant factors for a large percentage of antibiotics. In a recent study, Khan et al. [36] reported the existence of resistance carbapenemase genes, including NDM-1, KPC, and OXA-48 in *K. pneumoniae*. The collected outcomes of the current review indicate an association between antibiotic resistance and the presence of virulence genes.

### 3.3. Klebsiella Pneumoniae and COVID-19 Infections

COVID-19′s emergence attracted the attention of the whole world. Thus, all the medical studies have focused on it and on how to prevent its spread and ignored the accompanying infections. Studies concerning co-infection among COVID-19 patients have been somewhat few in number or neglected, particularly at the beginning of the outbreak of SARS-CoV-2. However, due to the conflicting interpretations of the mortality rate and the severity of COVID-19, it drew the attention of researchers, who aimed to search for the presence of co-infections. Different studies indicated that the outbreak of co-infections varied in the patients infected by SARS-CoV-2. Lai et al. [7] mentioned that a wide variation in co-infection rates was observed in patients infected with SARS-CoV-2 and extended from 0.6 to 50%. In fact, an expansive multicenter review investigation stated that for 50% of the diseased people who died of COVID-19, auxiliary bacterial co-infection occurred due to their being in hospital for treatment [37]. As a result of the studies on co-infection, it has become clear that the rate of viral–bacterial co-infection mortality is greater compared to that of bacterial or viral infection alone. This information was previously reported, where Brundage and Shanks [38] observed that viral infection promoted bacterial proliferation and increased the rate of infection. For example, the duplication of the influenza virus harmed the pneumonic virus, which exposes the connection positions designed for intrusion by bacteria and inactivates bacterial clearance from the breathing tract.

The problem of viral infections lies in the weakness of the immune system, which leads to the presence of secondary bacterial infections, as well as fungal infections. Consequently, as mentioned in earlier studies, bacterial infection complications are considered one of the leading causes of influenza-accompanied fatalities. Regarding immune system failure, Wang et al. [39] found that the lymphocytes, specifically the T, B, and NK cells, were harmed as a result of an infection caused by SARS-CoV-2. A decline in the lymphocytes and human immunity activity, according to Luo et al. [40], may be the greatest cause of co-infection. The treatment protocol for the SARS-CoV-2 infection includes corticosteroid compounds, as recommended by the World Health Organization. These compounds may increase the threat and severity of bacterial co-infection. The treatment also depends on the utilization of antibiotics in order to minimize the symptoms resulting from the damage of SARS-CoV-2. These antibiotics may increase the resistance of the bacteria to these compounds and thus delay the healing process.

Several randomized trials showed that treatment with corticosteroid compounds recovered the clinical outcomes and reduces the rate of mortality in hospitalized COVID-19 patients, particularly those who needed supplemental oxygen, but in the treatment of patients who did not require supplemental oxygen, corticosteroids reflected no improvement and, on the contrary, had harmful effects [41,42]. Generally, there are no documents to support the utilization of corticosteroids in non-hospitalized patients infected with SARS-CoV-2. Consequently, the treatment of SARS-CoV-2 diseases using corticosteroid compounds and antibiotics should be reconsidered.

Many species of bacteria and fungi were registered as co-pathogens among patients infected by COVID-19, such as *K. pneumoniae, Chlamydia pneumonia, S. pneumoniae, Mycoplasma pneumoniae, A. baumannii, S. aureus, L. pneumophila*, the *Candida* species, and the filamentous fungus *Aspergillus flavus*, in addition to certain co-pathogens, including metapneumovirus, influenza B virus, human immunodeficiency virus, coronavirus, enterovirus, and parainfluenza [7]. As observed in the present reports, COVID-19 patients were highly co-infected with bacteria compared with other microorganisms, and it was shown that the most common microbe was *S. pneumoniae*, followed by *K. pneumoniae*. In terms of expansion, the extent of viral, fungal, and bacterial–fungal co-infections remained the most elevated in serious COVID-19 cases [43]. Regarding *K. pneumoniae*, Montrucchio et al. [44] found that during treatment in the intensive care unit (ICU), carbapenemase-producing *K. pneumoniae* (CP-Kp) co-infected seven patients with COVID-19. A case study by Wei-Cheng et al. [45] observed the first mortality of a COVID-19 case in Taiwan, and it was known to be a result of extreme community-acquired pneumonia (CAP), *K. pneumoniae*, which was, at first, confined. Often, co-infections occur during the first four days of infection with the COVID-19 disease. In the United States, Massey et al. [46] explained that *K. pneumoniae* and *Moraxella catarrhalis* were the most common severe co-pathogens identified in SARS-CoV-2 patients, which can create severe respiratory infections. In a reference study in China, it was found that the second most common respiratory pathogen was *K. pneumoniae*, followed only *S. pneumoniae*, in patients with COVID-19 [43]. Italy has been one of the countries most affected by the COVID-19 epidemic, which has claimed many people’s lives. In Italian COVID-19 patients, Arcari et al. [47] noted that carbapenemase-producing *K. pneumoniae* was detected in COVID-19 patients in 34% of cases. In Saudi Arabia, *K. pneumoniae* and *A. baumannii* were the most prevalent bacterial species among 108 COVID-19 patients, showing complete resistance to all the tested antibiotics, except for colistin [48].

Said et al. [49] screened for microbial co-infection in 301 patients infected by SARS-CoV-2. Among the main detected microbes were multidrug-resistant *K. pneumoniae* (37%), followed by 26% of cases infected by extremely drug-resistant *A. baumannii*, 18.6% of cases infected by multidrug-resistant *E. coli*, 8.5% of cases infected by extremely drug-resistant *P. aeruginosa*, and 9.3% of cases infected with other bacteria. Moreover, it was observed that death rate increment was associated with bacterial infection, particularly with *K. pneumoniae* and *A. baumannii*. Moreover, Said et al. [49] mentioned that the co-infection percentage differed according to age, as summarized in Figure 4.

In another recent report, the rate of bacterial co-infections was confirmed in 1055 COVID-19 patients via polymerase chain reaction (PCR) tests, demonstrating that *K. pneumoniae* was among the other detected bacteria [50].

García-Meniño et al. [51] evaluated the effect of illnesses caused by multidrug-resistant *K. pneumoniae* in COVID-19-infected cases and recommended that the maximization of infection control measures is an important method that can be used to evade *K. pneumoniae* and other multidrug-resistant bacteria in COVID-infected patients. Similarity was observed in some symptoms of pneumonia caused by *K. pneumoniae* and SARS-Cov-2 (Figure 5).

One of the problems with the treatment of COVID-19 infections is the excessive use of antibiotics, which is a major cause of rise in antibiotic-resistant bacteria. Some *K. pneumoniae* strains are not only resistant to solitary antibiotics but to several antibiotics and, therefore, are said to be multi-drug resistant. Based on the greater percentage of antibiotic resistance, particularly colistin resistance, detected among the *K. pneumoniae* isolates, particularly in patients co-infected with COVID-19, it will be necessary to manage and evaluate the utilization of antibiotics continually.

A recent study [52] assumed that the greatest driver of the increment in antimicrobial resistance throughout the COVID-19 pandemic was the utilization of broad-spectrum antibiotics, in addition to a broad expanse of resistant microbes, such as carbapenem-resistant *K. pneumoniae* (CR-Kp), at the San Martino Policlinico Hospital IRCCS in Genoa, Italy. Why do prescriptions for COVID-19 infection treatment contain many broad-spectrum antibiotics? Many scientists have answered this question, as it may be due to the complexity involved in distinguishing between pneumonia bacterial co-infection and viral pneumonia infection alone [53,54]. Despite this, Hazra et al.’s [55] study assumed that infection with other respiratory pathogens is not common among COVID-19 patients. At the same time, Hazra et al. [55] mentioned that there are rather extraordinary differences in the number and sorts of co-infecting respiratory pathogens in COVID-19 patients across nations and locales. Therefore, according to the latest recommendation [56], it is important to pay attention to critical COVID-19 patients with bacterial co-infections. Thus, it is vital to consider bacterial co-infections in ordinary patients who are positive for COVID-19 [56].

However, at the beginning of the COVID-19 outbreak, there were no treatments that eliminated the virus, which subjected the whole world to great trouble. It later became a great challenge for many scientists to find an effective drug to solve this problem. For example, Wen et al. [57] studied the impacts of three novel drugs, namely fluvoxamine, molnupiravir, and Paxlovid, on COVID-19 treatment, and they observed that the mortality and hospitalization rates were reduced in patients infected with COVID-19. Furthermore, the three antiviral drugs did not increase the incidence of adverse events. Recently, the interaction between the S protein of coronavirus and the human cell ACE2 receptor was found to be inhibited by three compounds, TS-1276 (anthraquinone), tannic acid, and TS-984 (9-methoxycanthin-6-one), as mentioned in the study of Li et al. [58].

Although real-time reverse-transcription polymerase chain reaction (real-time RT-PCR) is the gold standard method for the identification of the causative agent of COVID-19 sickness, there are some factors that appear to be defective and affect the accuracy of this analytical test, such as the mutation that occurs in the virus, particularly in the target regions, and PCR inhibition, in addition cross-impurity between samples. Presently, the diagnostics features of COVID-19 have a critical function at the level of clinical administration and also for public health control and the increase in probable therapy options, as documented by Capalbo et al. [59]

In the current review, it is relevant to note that *K. pneumoniae* is biologically important due to its history of resistance to antibiotics as well as nosocomial/hospital infections. Some techniques have been used for COVID-19 and *K. pneumoniae* diagnosis (Figure 6).

## 4. Conclusions

The discrepancy in the rates of co-infection recurrence based on the recently distributed reports may be somewhat clarified by the seasonal and geographic changeability of respiratory pathogens. The initial discovery and protection of patients infected by SARS-CoV-2, particularly in the ITU, are critically required for the security measures of health-care laborers and in-hospital patients. During the outbreak of the COVID-19 pandemic, antibiotic resistance levels were high, and the commonly reported antibiotic resistant pathogen was *K. pneumonia*.

In general, it is imperative to limit the hazards of the disease and the outbreak of *K. pneumoniae* strains that are resistant to numerous antibiotics, which increase the risk of COVID-19 infection. The presence of co-infections must be taken into account when diagnosing COVID-19. In addition, the type and diversity of co-infected microbes should be emphasized. Moreover, the search for the causes of infection by co-infected microbes must attract the attention of scientists. Providers of healthcare, as well as people in the public, need a high awareness with respect to the suitable uses of antibiotics, both during pandemics and in normal situations. All the outcomes of the current review have important clinical implications for the development of COVID-19 treatments, especially in geriatric and critical situation management. Finally, urgent support from the authorities and policymakers is required in order to issue more limitations on the utilization of antibiotics, more so than in the present situation.

## Figures and Tables

**Figure 1 healthcare-10-02412-f001:**
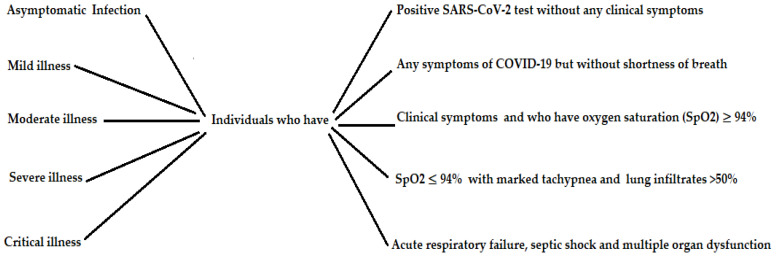
Severity levels of COVID-19.

**Figure 2 healthcare-10-02412-f002:**
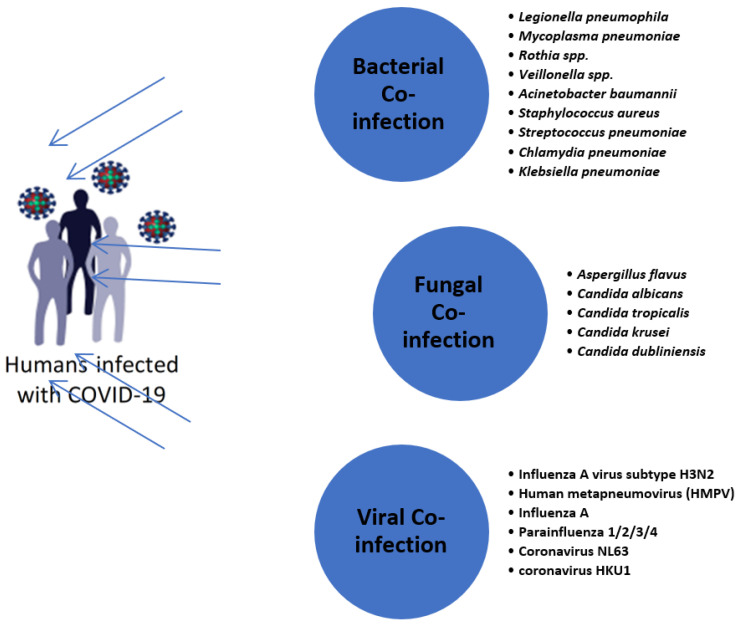
Co-infection by variable microbes among COVID-19 patients.

**Figure 3 healthcare-10-02412-f003:**
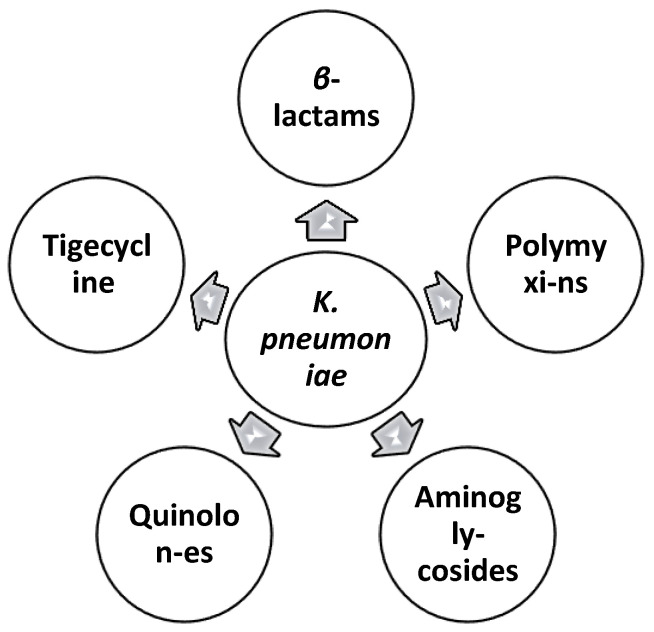
Resistance of *K. pneumoniae* to five important antibacterial classes.

**Figure 4 healthcare-10-02412-f004:**
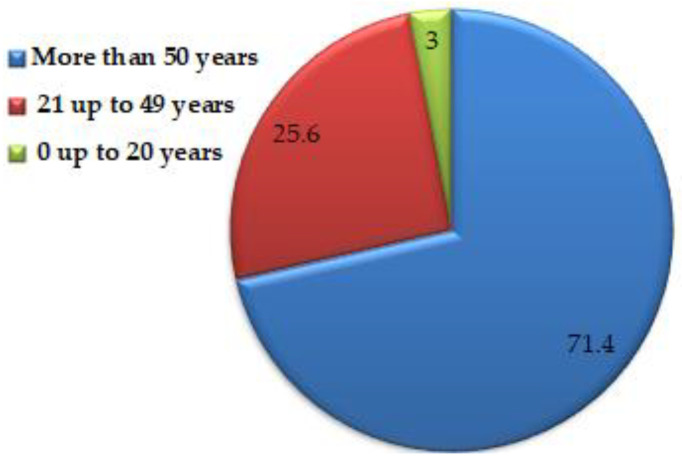
Co-infection % according to age.

**Figure 5 healthcare-10-02412-f005:**
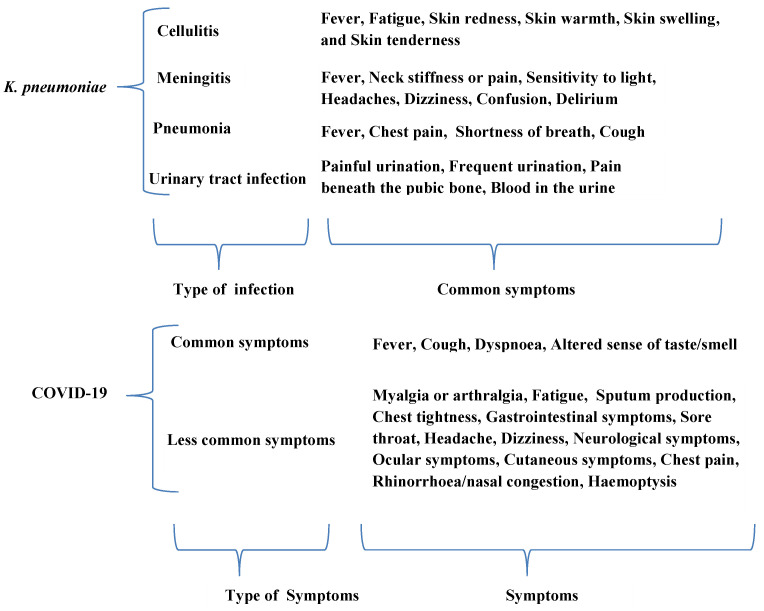
Schematic diagrams of *K. pneumoniae* and COVID-19 symptoms.

**Figure 6 healthcare-10-02412-f006:**
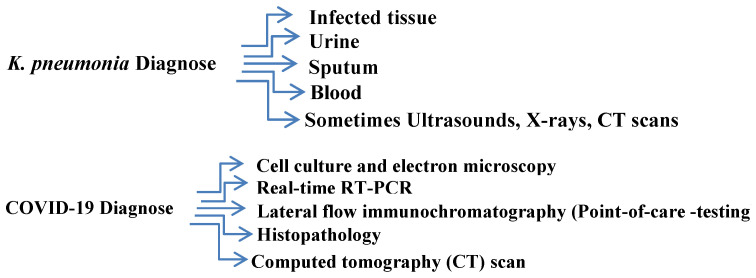
Schematic diagrams of *K. pneumoniae* and COVID-19 diagnosis.

## Data Availability

The datasets utilized during the present review are available from the corresponding author upon reasonable request.

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
