# Peer review of "Problems Associated with Co-Infection by Multidrug-Resistant Klebsiella pneumoniae in COVID-19 Patients: A Review"

_healthcare, 2022, doi:10.3390/healthcare10122412_

Round 1
Reviewer 1 Report (New Reviewer)
This review provides a thorough picture of multidrug-resistant Klebsiella pneumoniae co-infection in COVID-19 patients (hospitalised and outpatients), however the author needs to address the following issue before it can be accepted for publication.
Query follows
1. “To date, insufficient considers have examined bacterial co-infections in COVID-19 patients.Therefore, the present review outlined the current survey on K. pneumoniae co-infection to COVID-19 patients”. Line 94-97. Whether or not the author covered data from a particular area.? It is preferable to outline the area that the current investigation covers in the title, abstract and the body of the manuscript.
Author Response
“To date, insufficient considers have examined bacterial co-infections in COVID-19 patients.Therefore, the present review outlined the current survey on K. pneumoniae co-infection to COVID-19 patients”. Line 94-97. Whether or not the author covered data from a particular area.? It is preferable to outline the area that the current investigation covers in the title, abstract and the body of the manuscript.
In Saudi Arabia as well as other countries.

Reviewer 2 Report (New Reviewer)
It was really difficult to understand the framework of your study.
The introduction was long-winded.
The results and the conclusion were not clear.
Author Response
- Comments and Suggestions for Authors
It was really difficult to understand the framework of your study. It is your problem
The introduction was long-winded. Introduction according to previous reviewers
The results and the conclusion were not clear. There is Collected data with its discussion
Reviewer 3 Report (New Reviewer)
The title should include only the "Co-infection by multidrug-resistant Klebsiella pneumoniae in COVID-19 Patients" as there's no discussion of Hospitalised and non Hospitalised patients in the manuscript.
Citation is needed in some instances, line # 27, 37, 48, etc.
line#48 the sentence is needed to be rephrased "Carl 48 Friedlander in 1882, depicted Klebsiella pneumoniae in the primary interval"???
line#59 In spite of the fact that the full world work is difficult to understand???
line#85 Klebsiella species is located ???
line#105 Name the databases which were searched and which duration was selected for searching.
need appropriate citation "Moreover, who observed that death rates increment 325 was associated with bacterial infection particularly with K. pneumoniae and A. baumannii. Also, who mentioned that Co-infections % were differed according to age as summarized in fig. 4"
Is there a need to mention the outcome of one study in fig?? Figure 4 is not cited.
There are several grammatical mistakes, citations are needed in some places and the whole manuscript is needed to be thoroughly proofread.
Author Response
The title should include only the "Co-infection by multidrug-resistant Klebsiella pneumoniae in COVID-19 Patients" as there's no discussion of Hospitalised and non Hospitalised patients in the manuscript. The title changed to : Co-infection by multidrug-resistant Klebsiella pneumoniae in COVID-19 Patients
Citation is needed in some instances, line # 27, 37, 48, etc.
Abdelghany, T.M., Ganash, M., Bakri, M.M., Qanash, H., Al-Rajhi, A.M.H., and Elhussieny, N.I. SARS-CoV-2, the other face to SARS-CoV and MERS-CoV: future predictions. Biomedical Journal. 2021; 44:86–93.
World Health Organization
Sakurai A, Sasaki T, Kato S, Hayashi M, Tsuzuki SI, Ishihara T, Iwata M, Morise Z, Doi Y. Natural History of Asymptomatic SARS-CoV-2 Infection. N Engl J Med. 2020 Aug 27;383(9):885-886. doi: 10.1056/NEJMc2013020.
line#48 the sentence is needed to be rephrased "Carl 48 Friedlander in 1882, depicted Klebsiella pneumoniae in the primary interval"??? Klebsiella pneumoniae was isolated from respiratory system of humans infected with pneumonia, and examined by Carl Friedlander in 1882, it appeared in bacillus form.
line#59 In spite of the fact that the full world work is difficult to understand???
In spite of the fact that the full world work to solute the outbreak of COVID-19, the fitting management and immunization continue vague around COVID-19 widespread.
line#85 Klebsiella species is located ???
According to earlier studies, Klebsiella species was consider among the highest
line#105 Name the databases which were searched and which duration was selected for searching.
Also, the databases on Web of Science were…
need appropriate citation "Moreover, who observed that death rates increment 325 was associated with bacterial infection particularly with K. pneumoniae and A. baumannii. Also, who mentioned that Co-infections % were differed according to age as summarized in fig. 4"
Said KB, Alsolami A, Moussa S, Alfouzan F, Bashir AI, Rashidi M, Aborans R, Taha TE, Almansour H, Alazmi M, Al-Otaibi A, Aljaloud L, Al-Anazi B, Mohialdin A, Aljadani A. COVID-19 Clinical Profiles and Fatality Rates in Hospitalized Patients Reveal Case Aggravation and Selective Co-Infection by Limited Gram-Negative Bacteria. Int J Environ Res Public Health. 2022 Apr 26;19(9):5270. doi: 10.3390/ijerph19095270.
Is there a need to mention the outcome of one study in fig?? Figure 4 is not cited.
Its present in line 327 fig. 4
There are several grammatical mistakes, citations are needed in some places and the whole manuscript is needed to be thoroughly proofread.
Its corrected

Reviewer 4 Report (New Reviewer)
The authors have submitted a study which is truly of interest.
There are a few comments
ENGLISH needs extensive attention.
The uploaded file is marked by different colours. The authors should upload a clean file.
Abstract
The beginning of the abstract is a confusing statement. please revise and make it straightforward understanding.
This is a review and not a "review study" as mentioned by the authors. Please revise.
The abstract should end with the aim if the review. Its missing please add.
SARS-CoV-2 is the 7th CoV identified to infect humans. As known it belongs to the subgenus Sarbecovirus of the Coronaviridae family..... these statements need references.
The placement of figure 1 should be earlier than where it is placed currently.
According to available data from WHO, from the beginning of COVID-19 outbreaks till 31 July 2022 there were nearly more than 574 million confirmed cases and nearly more than 6.3 million deaths associated with SARS-CoV-2 and its variants worldwide... From where this data was retrieved?
Figure 3. Please make it more appealing its plain with gross font errors.
Please follow the guidelines while citing the Figures. The references are not as per the guidelines of the journal.
Author Response
There are a few comments
ENGLISH needs extensive attention. Its corrected
The uploaded file is marked by different colours. The authors should upload a clean file.
According to previous corrections , Its corrected
Abstract
The beginning of the abstract is a confusing statement. please revise and make it straightforward understanding.
This is a review and not a "review study" as mentioned by the authors. Please revise.
Its corrected to a review
The abstract should end with the aim if the review. Its missing please add.
Therefore, the impact of K. pneumoniae co-infection with COVID-19 patients was the aim of present review.
SARS-CoV-2 is the 7th CoV identified to infect humans. As known it belongs to the subgenus Sarbecovirus of the Coronaviridae family..... these statements need references.
Abdelghany, T.M., Ganash, M., Bakri, M.M., Qanash, H., Al-Rajhi, A.M.H., and Elhussieny, N.I. SARS-CoV-2, the other face to SARS-CoV and MERS-CoV: future predictions. Biomedical Journal. 2021; 44:86–93.
The placement of figure 1 should be earlier than where it is placed currently.
Its transferred
According to available data from WHO, from the beginning of COVID-19 outbreaks till 31 July 2022 there were nearly more than 574 million confirmed cases and nearly more than 6.3 million deaths associated with SARS-CoV-2 and its variants worldwide... From where this data was retrieved?
World Health Organization
Figure 3. Please make it more appealing its plain with gross font errors.
Its changed
Please follow the guidelines while citing the Figures. The references are not as per the guidelines of the journal.
Ok if paper accepted the references will rewrite
Round 2
Reviewer 4 Report (New Reviewer)
Please perform an extensive evaluation of the written manuscript yet again and check for errors.
please find some examples below
"The impact of co-infection by K. pneumoniae in COVID-19 patients has yet to be estimated in this a review study. " the sentence is confusing please check
The greatest studies identify and recorded...... "greatest"??
There is similar among SARS-CoV-2 and other previous detected coronavirus (SARS-CoV and MERS-CoV), but it is distinct from them [1]. It is unclear what the authors want to say here?
"medicinal reports " ???
According to earlier studies, Klebsiella species was conceded among the highest...... surprisingly the authors have cited only one paper. Please revise.
Please do a valid English proofreading throughout.
Author Response
Dear reviewer
Very thanhs
please se the responses with green colour in manuscript text
Comment : Please perform an extensive evaluation of the written manuscript yet again and check for errors.
Response : its checked
please find some examples below
"The impact of co-infection by K. pneumoniae in COVID-19 patients has yet to be estimated in this a review study. " the sentence is confusing please check
Response : its changed to: The impact of co-infection by K. pneumoniae in COVID-19 patients has been estimated in the current a review study.
The greatest studies identify and recorded...... "greatest"??
Response : its changed to: The spread of K. pneumoniae as a co-infection between critically ill COVID-19 patients particularly throughout hospitalization was identified and recorded via numerous reports
There is similar among SARS-CoV-2 and other previous detected coronavirus (SARS-CoV and MERS-CoV), but it is distinct from them [1]. It is unclear what the authors want to say here?
Response : its corrected to: There is some similarities among SARS-CoV-2 and other previous detected coronavirus (SARS-CoV and MERS-CoV), but it is distinct from them in some things such as transmission rate, origin, mortality rate [1].
"medicinal reports " ???
Response : its changed to published medicinal reports
According to earlier studies, Klebsiella species was conceded among the highest...... surprisingly the authors have cited only one paper. Please revise.
Response : its corrected to: species was considered among the…..
Please do a valid English proofreading throughout.
Response : It is ok

This manuscript is a resubmission of an earlier submission. The following is a list of the peer review reports and author responses from that submission.
Round 1
Reviewer 1 Report
In this study R. Yahya reviews the role of MDR K. Pneumoniae co-infection in COVID-19 patients. In the review the author gives an overview of K.Pneumoniae with it's virulence traits, an historical evaluation of antimicrobial resistance of this bacteria and a short evaluation of SARS-CoV-2 and MDR K.Pneumonae co-infection.
Firstly I need to underlight the need to edit this paper with an expert English writer, becouse the paper is hard to read end with several English errors or misleading words. For this reason I suggest to rewrite the paper and resubmit it after this editing.
Furthermore, some other issues about the paper that should be evluated to increase it's scientific interest are:
- The focus should be on the central theme of the paper: K.Pneumoniae and SARS-CoV-2 co-infection, with more citations of studies, that should be updated to 2021. The section is shorter then expexted, and evidence on this theme is rising, compared to the cited literature (e.g. 10.3390/antibiotics11070894.; doi:10.2147/IJGM.S359959.; doi:10.3390/ijerph19095270.; doi:10.7759/cureus.22363.; doi: 10.3390/microorganisms10030495.) , even if the problem was neglected at the start of the pandemic, as stated by the author. There are also new studies that are focused on explaining the possible role of the virus in this co-infection: doi:10.3389/fimmu.2022.841759.
- In figure 1 the asymptomatic population stated by the author is 1%, however even the first epidemiological studies, such as Lavezzo et al. (https://doi.org/10.1038/s41586-020-2488-1) or Gudbjartsson et al. (DOI: 10.1056/NEJMoa2006100) found an asymptomatic percentage of nearly 40%, data confirmed in a recent methanalysis (Ma et al. doi: 10.1001/jamanetworkopen.2021.37257.).
- the section regarding K.Pneumoniae spectrum antibiotic resistance is somehow confusing, I would prefer a more concise paragraph, more divided on the basis of resistance mechanism and not on the historical appereance of the resistance.
-The author states (page 6 line 220-221) that the steroid treatment in COVD-19 should be reconsidered, however, even with the problematic associated, it has a central role in the COVID-19 treatment and is one of the few treatments with scientifc proven efficacy.
- In line 268 the author states that there are no antiviral treatment for SARS-CoV-2, however some new drugs have been approved for COVID-19 early treatment (doi: 10.1080/07853890.2022.2034936.). The statement should be updated.
Author Response
Dear my prof. very thanks for reviewing my manuscript
Comment: Firstly I need to underlight the need to edit this paper with an expert English writer, because the paper is hard to read end with several English errors or misleading words
Response: Its edited with yellow highlights
Comment: The focus should be on the central theme of the paper: K.Pneumoniae and SARS-CoV-2 co-infection, with more citations of studies, that should be updated to 2021. The section is shorter then expexted, and evidence on this theme is rising, compared to the cited literature (e.g. 10.3390/antibiotics11070894.; doi:10.2147/IJGM.S359959.; doi:10.3390/ijerph19095270.; doi:10.7759/cureus.22363.; doi: 10.3390/microorganisms10030495.) , even if the problem was neglected at the start of the pandemic, as stated by the author. There are also new studies that are focused on explaining the possible role of the virus in this co-infection: doi:10.3389/fimmu.2022.841759.
Response: all references and others were cited in my manuscript for improvement
Comment: In figure 1 the asymptomatic population stated by the author is 1%, however even the first epidemiological studies, such as Lavezzo et al. (https://doi.org/10.1038/s41586-020-2488-1) or Gudbjartsson et al. (DOI: 10.1056/NEJMoa2006100) found an asymptomatic percentage of nearly 40%, data confirmed in a recent methanalysis (Ma et al. doi: 10.1001/jamanetworkopen.2021.37257.).
Response: the following was added in manuscript: Based on the severity of the offering illness, COVID-19 was classified into five distinct kinds (Fig.1), however it was different among reports conducted in different locations, moreover there is still some doubt about the percentage of asymptomatic infections. Furthermore, the description of asymptomatic infections may differ within the medicinal reports, depending on which definite symptoms were judged. For instance according to Sakurai et al. [2] at the time of diagnosis, 58 % of the 712 individuals confirmed with COVID-19 were asymptomatic. Ma et al. [3] mentioned that the individuals with asymptomatic infections represent 40.50%. During the outbreak of Omicron variant, in South Africa a large report assessed that 31% of the confirmed cases were asymptomatic. Therefore, according to Ma et al. [3], discovering the asymptomatic infection is essential, particularly in regions that have effectively controlled SARS-CoV-2.
the following was cited : Ma Q, Liu J, Liu Q, Kang L, Liu R, Jing W, Wu Y, Liu M. Global Percentage of Asymptomatic SARS-CoV-2 Infections Among the Tested Population and Individuals With Confirmed COVID-19 Diagnosis: A Systematic Review and Meta-analysis. JAMA Netw Open. 2021 Dec 1;4(12):e2137257. doi: 10.1001/jamanetworkopen.2021.37257.
Comment: the section regarding K.Pneumoniae spectrum antibiotic resistance is somehow confusing, I would prefer a more concise paragraph, more divided on the basis of resistance mechanism and not on the historical appereance of the resistance.
Response: According to Du et al. [28], mechanism of tigecycline resistance in K. pneumoniae has still not been obviously clarified, however tetA gene may responsible for antibiotcs resistance . Therefore, Du et al. [28], confirmed that all clinical strains of K. pneumoniae that carry tetA are associated to the failure treatment by tigecycline. A number of resistance genes were recognized in seven strains of K. pneumoniae (three tigecycline-resistant and four colistin-resistant strains). Sequencing of the gene mutations [29], the sequencing of the gene mutations from the resistant K. pneumoniae strains, showed that mutations of ramR and lon genes were the probable response for tigecycline resistance while colistin resistance may due to pmrB, phoQ and mgrB genes. Non-specific active resistance-nodulation-cell division efflux pumps for instance AcrAB-TolC was associated as a mechanism with tigecycline-resistant K. pneumonia [30]. Other mechanisms to antibiotics resistance were elucidated in K. pneumoniae, for example modifications of β-lactam antibiotics binding proteins and permeability of the outer membrane. Point mutations Resistance to fluoroquinolone. As reported by Moya and Maicas [31] the modifications in the DNA gyrase at specific regions (gyrA and gyrB genes), and topoisomerase IV at specific regions (genes parC and parE) lead to resistance of fluoroquinolone.
the following was cited
- Du X, He F, Shi Q, Zhao F, Xu J, Fu Y, Yu Y. The Rapid Emergence of Tigecycline Resistance in blaKPC-2Harboring Klebsiella pneumoniae, as Mediated in Vivoby Mutation in tetA During Tigecycline Treatment. Front Microbiol. 2018 5;9:648. doi: 10.3389/fmicb.2018.00648.
- JinX, Chen Q, Shen F, Jiang Y, Wu X , Hua X, Fu Y, Yu Y. Resistance evolution of hypervirulent carbapenem-resistant Klebsiella pneumoniae ST11 during treatment with tigecycline and polymyxin. Emerg Microbes Infect, 2021 ,10(1):1129-1136. doi: 10.1080/22221751.2021.1937327.
30 Wang, X., Xu, X., Zhang, S., Chen N, Sun Y, Ma K, Hong D, Li L, Du Y, Lu X, Jiang S.TPGS-based and S-thanatin functionalized nanorods for overcoming drug resistance in Klebsiella pneumonia. Nat Commun 13, 3731 (2022). https://doi.org/10.1038/s41467-022-31500-3
31 Moya, C.; Maicas, S. Antimicrobial Resistance in Klebsiella pneumoniae Strains: Mechanisms and Outbreaks. Proceedings 2020, 66, 11. https://doi.org/10.3390/proceedings2020066011
Comment -The author states (page 6 line 220-221) that the steroid treatment in COVD-19 should be reconsidered, however, even with the problematic associated, it has a central role in the COVID-19 treatment and is one of the few treatments with scientifc proven efficacy.
Response: yes , it has a central role in the COVID-19 treatment. I added the following in manuscript regarding these subject: Several randomized trials showed that treatment by corticosteroid compounds recovers clinical outcomes and reduces the rate of mortality in hospitalized COVID-19 patients particularly who need supplemental oxygen, but the treatment of patients who do not require supplemental oxygen, corticosteroids reflected no any improvement, on the contrary, it has harmful effects (Li et al. 2021, Wagner et al. 2021). Generally, there are no documents to support the utilize of corticosteroids in nonhospitalized patients infected with SARS-CoV-2. Consequently, the treatment of SARS-CoV-2 diseases using corticosteroid compounds and antibiotics should be reconsidered.
Li H, Yan B, Gao R, Ren J, Yang J. Effectiveness of corticosteroids to treat severe COVID-19: A systematic review and meta-analysis of prospective studies. Int Immunopharmacol. 2021 Nov;100:108121. doi: 10.1016/j.intimp.2021.108121.
Wagner C, Griesel M, Mikolajewska A, Mueller A, Nothacker M, Kley K, Metzendorf MI, Fischer AL, Kopp M, Stegemann M, Skoetz N, Fichtner F. Systemic corticosteroids for the treatment of COVID-19. Cochrane Database Syst Rev. 2021 Aug 16;8(8):CD014963. doi: 10.1002/14651858.CD014963.
Comment - In line 268 the author states that there are no antiviral treatment for SARS-CoV-2, however some new drugs have been approved for COVID-19 early treatment (doi: 10.1080/07853890.2022.2034936.). The statement should be updated.
Response: its updated However at the beginning of the COVID-19 outbreak, there were no any treatments that eliminated the virus, which put the whole world in great trouble. Which later became a great challenge for many scientists to find an effective drugs to solve this problem. For example, Wen et al.[2022] studied the impact of three novel drugs namely fluvoxamine, molnupiravir and Paxlovid on COVID-19 treatment, who observed that the mortality and hospitalization rates were reduced in patients infected with COVID-19, furthermore the three antiviral drugs did not rise the incidence of adverse events. Recently, the interaction between the S protein of coronavirus and the human cell ACE2 receptor was inhibited by three compounds TS-1276 (anthraquinone) tannic acid, and TS-984 (9-Methoxycanthin-6-one) as mentioned in study of Li et al. (2022).
Wen Wen, Chen Chen, Jiake Tang, Chunyi Wang, Mengyun Zhou, Yongran Cheng, Xiang Zhou, Qi Wu, Xingwei Zhang, Zhanhui Feng, Mingwei Wang & Qin Mao (2022) Efficacy and safety of three new oral antiviral treatment (molnupiravir, fluvoxamine and Paxlovid) for COVID-19:a meta-analysis, Annals of Medicine, 54:1, 516-523, DOI: 10.1080/07853890.2022.2034936
Li, C., Zhou, H., Guo, L. et al. Potential inhibitors for blocking the interaction of the coronavirus SARS-CoV-2 spike protein and its host cell receptor ACE2. J Transl Med 20, 314 (2022). https://doi.org/10.1186/s12967-022-03501-9
Santoso P, Sung M, Hartantri Y, Andriyoko B, Sugianli AK, Alisjahbana B, Tjiam JSL, Debora J, Kusumawati D, Soeroto AY. MDR Pathogens Organisms as Risk Factor of Mortality in Secondary Pulmonary Bacterial Infections Among COVID-19 Patients: Observational Studies in Two Referral Hospitals in West Java, Indonesia. Int J Gen Med. 2022 May 6;15:4741-4751. doi: 10.2147/IJGM.S359959.
Said KB, Alsolami A, Moussa S, Alfouzan F, Bashir AI, Rashidi M, Aborans R, Taha TE, Almansour H, Alazmi M, Al-Otaibi A, Aljaloud L, Al-Anazi B, Mohialdin A, Aljadani A. COVID-19 Clinical Profiles and Fatality Rates in Hospitalized Patients Reveal Case Aggravation and Selective Co-Infection by Limited Gram-Negative Bacteria. Int J Environ Res Public Health. 2022 Apr 26;19(9):5270. doi: 10.3390/ijerph19095270.
Bahceci I, Yildiz IE, Duran OF, Soztanaci US, Kirdi Harbawi Z, Senol FF, Demiral G. Secondary Bacterial Infection Rates Among Patients With COVID-19. Cureus. 2022 Feb 18;14(2):e22363. doi: 10.7759/cureus.22363.

Reviewer 2 Report
First of all I would like to thank for the opportunity to review this paper. COVID-19 is an ongoing pandemic that has resulted in global health, economic and social crises. The importance of studying bacterial co-infections and antimicrobial susceptibility, serotype distribution of the main microorganism associated to invasive diseases is well known and emphasized due to the importance of their impact also on healthcare associated infection control. In this context, aim of the paper seems to review the impact of co-infection by K. pneumoniae in COVID-19 patients and the caratheristics of the coinfections.
The subject under study is certainly important, especially in the historical period we are experiencing. The article presents some interesting results, but it is nevertheless believed that, given the methodology used, the manuscript is not suitable for publication in the present form. I hope that these comments will not discourage the authors and that they will take them as an impulse to improve.
Title: it must be improved highlight the object of the study: place, time and person.
Introduction: The authors should improve the introduction, making clearer what is the gap in the literature that is filled with this study. The authors must better frame their study within the vast body of literature that addressed also the issue of related costs of healthcare associate infections (refer to articles with DOI: 10.2174/1389201020666190408095811)
Methods: This section is completely missing: the study is not a systematic review because it also included review papers. It is not a narrative review because it included quantitative analysis. It is not an umbrella review because it also included original studies. It is not a scoping review because it tested a specific research question and pooled quantitative data. What kind of review is it? How were the paper selected? What were the criteria of inclusion? All these issues must be addressed and specified.
Discussion: the discussion need to be expanded. The Authors do not fully emphasize the contribution of the study to the literature. The discussion is not updated in light of the economic impact of these infections (see the above mentioned reference). The Authors should add more practical recommendations for the reader, based on their findings.
Author Response
Comments and Suggestions for Authors
Comment: First of all I would like to thank for the opportunity to review this paper. COVID-19 is an ongoing pandemic that has resulted in global health, economic and social crises. The importance of studying bacterial co-infections and antimicrobial susceptibility, serotype distribution of the main microorganism associated to invasive diseases is well known and emphasized due to the importance of their impact also on healthcare associated infection control. In this context, aim of the paper seems to review the impact of co-infection by K. pneumoniae in COVID-19 patients and the caratheristics of the coinfections. The subject under study is certainly important, especially in the historical period we are experiencing. The article presents some interesting results, but it is nevertheless believed that, given the methodology used, the manuscript is not suitable for publication in the present form. I hope that these comments will not discourage the authors and that they will take them as an impulse to improve.
Response: Very thanks for encouragement
Comment: Title: it must be improved highlight the object of the study: place, time and person.
Response: changed to : Co-infection by multidrug-resistant Klebsiella pneumoniae in Hospitalised and non Hospitalised COVID-19 Patients
Comment: Introduction: The authors should improve the introduction, making clearer what is the gap in the literature that is filled with this study. The authors must better frame their study within the vast body of literature that addressed also the issue of related costs of healthcare associate infections (refer to articles with DOI: 10.2174/1389201020666190408095811)
Response: DOI: 10.2174/1389201020666190408095811 was used and other updated references
Comment: Methods: This section is completely missing: the study is not a systematic review because it also included review papers. It is not a narrative review because it included quantitative analysis. It is not an umbrella review because it also included original studies. It is not a scoping review because it tested a specific research question and pooled quantitative data. What kind of review is it? How were the paper selected? What were the criteria of inclusion? All these issues must be addressed and specified.
Response: The used search terms
The collected data in the current review were dependent on the several scientific papers written in English that available on MEDLINE website via the search terms such as medico-legal litigation COVID-19, liability claims COVID-19, medical liability pandemic and malpractice SARS-CoV-2. The goal of the present review is to focus on the documents, articles, reports and study cases that contain useful information on the virulence factors of K. pneumoniae, co-infections by Klebsiella pneumoniae in Covid-19 infections.
Comment: Discussion: the discussion need to be expanded. The Authors do not fully emphasize the contribution of the study to the literature. The discussion is not updated in light of the economic impact of these infections (see the above mentioned reference). The Authors should add more practical recommendations for the reader, based on their findings.
Response: several references were cited in the current review
Giraldi G, Montesano M, Napoli C, Frati P, La Russa R, Santurro A, Scopetti M, Orsi GB. Healthcare-Associated Infections Due to Multidrug-Resistant Organisms: a Surveillance Study on Extra Hospital Stay and Direct Costs. Curr Pharm Biotechnol. 2019;20(8):643-652. doi: 10.2174/1389201020666190408095811. PMID: 30961489.

Round 2
Reviewer 1 Report
Firtly I'd like to thank the author for the work made on the paper, that partially responds my doubts.
However, I belive that the paper continues to need an extensive English review, and i suggest the author to take some time with an English mothertongue reviewer to perform this task. So I suggest the author to re submit the work only after this work and I'm sorry to suggest the editor to reject the work for now.
Furthermore, I belive that the author has responded only partially to my second comment "the section regarding K.Pneumoniae spectrum antibiotic resistance is somehow confusing, I would prefer a more concise paragraph, more divided on the basis of resistance mechanism and not on the historical appereance of the resistance.". I suggest to review completly this paragraph, reducing the lenght and making it more schematic and less historical.
Reviewer 2 Report
The paper was greatly improved and it is now suitable for publication